# Association between type 2 diabetes and osteoporosis risk: A representative cohort study in Taiwan

**Hsin-Hui Lin**[1], **Hsin-Yin Hsu**[2,3], **Ming-Chieh Tsai**[2,4,5,6], **Le-Yin Hsu**[2], **Kuo-Liong Chien**[2,7], **Tzu-Lin Yeh**[2,8]*

**1** Department of Family Medicine, MacKay Memorial Hospital, Tamsui Branch, New Taipei City, Taiwan,
**2** Institute of Epidemiology and Preventive Medicine, National Taiwan University, Taipei, Taiwan,
**3** Department of Family Medicine, MacKay Memorial Hospital, Taipei, Taiwan, **4** Department of Endocrinology, Mackay Memorial Hospital, Tamsui Branch, New Taipei City, Taiwan, **5** Department of Internal Medicine, Mackay Memorial Hospital, Tamsui Branch, New Taipei City, Taiwan, **6** Department of Medical Research, Mackay Memorial Hospital, Taipei, Taiwan, **7** Department of Internal Medicine, National Taiwan University Hospital, Taipei, Taiwan, **8** Department of Family Medicine, Hsinchu MacKay Memorial Hospital, Hsinchu City, Taiwan

* 5767@mmh.org.tw

**Data Availability Statement:** The datasets generated and analyzed herein are not publicly available due to the terms of consent to which the participants agreed. However, such data can be

## Abstract

Although previous studies have investigated the relationship between fracture risk and type 2 diabetes (T2D), cohort studies that estimate composite osteoporosis risk are lacking. This retrospective cohort study sought to determine the risk of osteoporosis in Taiwanese patients with T2D. Patients diagnosed with T2D between 2002 and 2015 identified through the 2002 Taiwan Survey of Hypertension, Hyperglycemia, and Hyperlipidemia were included. A total of 1690 men and 1641 women aged ≥40 years linked to the National Health Insurance Research Database (NHIRD) were followed up to the end of 2015 to identify the incidences of osteoporosis through ICD9-CM codes for osteoporosis or osteoporotic fractures or usage of anti-osteoporotic agents according to Anatomical Therapeutic Chemical codes determined from NHIRD. The person year approach and Kaplan–Meier analysis were then used to estimate the incidences and cumulative event rates, whereas the Cox proportional hazard model was used to calculate adjusted hazard ratios (HR) for osteoporosis events. A total of 792 new osteoporosis events were documented over a median follow-up duration of 13.6 years. Participants with T2D had higher osteoporosis risk [adjusted HR: 1.37, 95% confidence interval (CI): 1.11–1.69] compared with those without T2D. Subgroup analyses revealed that age had a marginally significant effect, indicating that T2D had a more pronounced effect on osteoporosis risk in younger population (<65 years old). No difference was found between patients stratified according to sex. In conclusion, T2D was significantly associated with increased osteoporosis risk, especially in younger participants.

accessed from the authors upon reasonable request and with permission from the Health Promotion Administration at the Ministry of Health and Welfare in Taiwan at https://dep.mohw.gov.tw/DOS/cp-2516-3591-113.html (approved number: H108044). All interested researchers can direct their data inquiries to ntuepm@ntu.edu.tw, an institutional, non-author point of contact. Data are also available from the National Taiwan University Hospital Institutional Data Access/Ethics Committee (contact via ntuepm@ntu.edu.tw) for researchers who satisfy the criteria for accessing confidential data.

**Funding:** The authors received no specific funding for this work.

**Competing interests:** The authors have declared that no competing interests exist.

## Introduction

Osteoporosis, one of the most common metabolic diseases, is characterized by low mineral bone mass and microarchitectural deterioration of bone tissue, which consequently increases bone fragility and susceptibility to fractures [1]. Osteoporosis can occur in both men as well as in women, particularly aged [2, 3], and it is a significant health burden due to its high morbidity, mortality, and healthcare costs. A nutrition and health survey in Taiwan from 2013 to 2015 revealed an overall prevalence of osteoporosis at 12.3% [4]. Several risk factors for osteoporosis have been identified, some of which are complex due to the multiple mechanisms involved, such as type 2 diabetes (T2D).

T2D is also a considerably prevalent health burden both in Taiwan and worldwide [5]. According to the Nutrition and Health Survey, the overall prevalence of T2D in Taiwan is 9.3% diabetes [6]. T2D affects sugar, fat, and protein metabolism while also causing dysregulation of calcium, phosphorus, and magnesium, and subsequently promoting a series of complications, such as neuropathy, and cardiovascular, peripheral vascular, retinal, and metabolic bone diseases [7–9].

The effects of diabetes on the bone are complex. Although most studies agree that diabetes increases fracture risk [10–14], the association between T2D and risk for bone mineral density (BMD) loss have been inconsistent in prior studies, with abundant data showing a greater baseline BMD in individuals with T2D [15–17]. Osteoporosis is clinically diagnosed using both the World Health Organization criteria based on BMD [18] and the incidence of fragility fractures without T-score data [19]. Although numerous studies have assessed the risk of fractures or BMD change, only a few cross-sectional studies have explored the composite risk of osteoporosis in diabetes [20].

Given the growing prevalence of T2D and osteoporosis as well as the lack of information from cohort-based studies, this study aimed to determine the relationship between T2D and osteoporosis among a Taiwanese population.

## Materials and methods

### Study design and participants

This nationwide, representative community-based cohort study included all patients aged $\geq$ 40 years from the 2002 Taiwanese Survey on Prevalence of Hypertension, Hyperglycemia, and Hyperlipidemia (TwSHHH). The exclusion criteria were osteoporosis diagnosis, use of medication for osteoporosis, and a history of non-traumatic fractures diagnosed on the basis of International Classification of Diseases, Ninth Revision, Clinical Modification (ICD-9-CM) codes (S1 Table) at least twice as per the outpatient department note or once as per the hospitalization discharge note before the index date. Those who were pregnant within 1 year of the 2002 TwSHHH and those with type 1 diabetes diagnosed using ICD-9-CM codes (250 ×1 and 250 × 3) at least twice as per the outpatient department note or once as per hospitalization discharge note were also excluded.

The 2002 TwSHHH followed the 2001 National Health Interview Survey (NHIS) of Taiwan, a nationwide health survey, between March 2002 and October 2002 [21]. Half of the primary sampling units of the 2001 NHIS were randomly selected, and all members aged $\geq$15 years were interviewed for the 2002 TwSHHH study [22].

All participants of the 2002 TwSHHH were interviewed using a questionnaire that include questions on sociodemographic characteristics, dietary characteristics, menopause status in women, and personal histories, including smoking, exercise status, medical history (hypertension, diabetes, etc.), medication history, and family history of cardiovascular diseases.

Moreover, participants underwent physical examinations, to measure body mass index (BMI), waist and hip circumferences, and blood pressure (BP), and laboratory tests to determine parameters such as fasting plasma glucose (FPG), hemoglobin A1c (HbA1c), uric acid, renal and liver function, and lipid profiles. All laboratory data were obtained using standard protocols. The coefficients of variations for these laboratory measurements were approximately 5%. A total of 7578 participants in the 2002 TwSHHH completed the questionnaires, and 6941 of them had BP data (634 refusals) and 6602 provided a blood sample. Finally, 6600 participants completed the survey, and the data of 5786 participants (2927 men and 2959 women) were successfully linked with their data from the 2001 NHIS database [23].

Taiwan's National Health Insurance (NHI) program, which covers >99% of its 23 million residents [24], is a mandatory single-payer medical insurance program that was implemented in 1995 and it provides comprehensive health care services. The National Health Insurance Research Database (NHIRD) includes all data on NHI resource utilization, including outpatient visits, hospital care, prescribed medications, and National Death Registry. All data in the NHIRD are anonymous and encrypted for security purposes. Researchers using the NHIRD must state that they have no any intention of violating the insurants' privacy. The data of the 2002 TwSHHH populations are linked with that of the 2001 NIHS and NHIRD.

The datasets generated and analyzed herein are not publicly available due to the terms of consent to which the participants agreed. However, such data can be accessed from the authors upon reasonable request and with permission from the Health Promotion Administration at the Ministry of Health and Welfare in Taiwan at https://dep.mohw.gov.tw/DOS/cp-2516-3591-113.html (approved number: H108044). All interested researchers can direct their data inquiries to ntuepm@ntu.edu.tw, an institutional, non-author point of contact. Data are also available from the National Taiwan University Hospital Institutional Data Access/Ethics Committee (contact via ntuepm@ntu.edu.tw) for researchers who satisfy the criteria for accessing confidential data.

Our protocol was reviewed and approved by the Research Ethics Committee of National Taiwan University Hospital. The committee was organized under and operates in accordance with the Good Clinical Practice Guidelines [NTUH-REC Number: 201901103W (Institutional Review Board reference)]. Written informed consent was obtained from all participants.

## Definition of diabetes

Diabetes was defined as an FPG concentration of $\geq$ 126 mg/dL or HbA1c of $\geq$ 6.5% determined from the 2002 TwSHHH or as the use of antidiabetics according to the Anatomical Therapeutic Chemical (ATC) codes determined from the NHIRD ($\geq$ 28 tablets 1 year before the index date) (S2 Table).

## Definition of outcome

This study used a composite diagnosis of osteoporosis comprising at least two outpatient diagnosis or one hospital discharge diagnosis based on ICD9-CM codes for osteoporosis or non-traumatic fractures (S1 Table) or the use of $\geq$14 tablets or one injectable dose of anti-osteoporotic agents (excluding calcium, vitamin D, or hormone replace therapy) between 2002 and 2015 according to the ATC codes (S2 Table) determined from the NHIRD.

## Covariates

Patient characteristics, including age, sex, menopause status, calcium consumption status, exercise habits, smoking status, daily alcohol consumption, and socioeconomic status (including marital status, income, and education level) were obtained from the 2001 NHIS and 2002

TwSHHH. Medication- and diagnosis based comorbidities were determined from the NHIRD (S3 Table).

## Statistical analyses

Descriptive analyses were performed. Categorical and continuous variables were analyzed using the chi-square test and analysis of variance, respectively. Kaplan–Meier survival curves for osteoporosis events were plotted according to baseline diabetes status, after which the log rank test was performed to determine the differences between both groups. Person-years were calculated for each participant from the date at which the 2002 TwSHHH questionnaire was competed until the occurrence of an event, death, or till December 31, 2015, whichever came first. Osteoporosis incidence rates were expressed as the number of cases divided by the number of 1000-person years of follow-up.

Cox proportional hazards regression analysis was performed to determine multivariable adjusted hazard ratios (HRs) and 95% confidence intervals (CIs) after testing the proportionality assumption using the time-dependent covariate method [25]. Potential confounders were then adjusted using three models. Baseline model 1 adjusted for sex (male, female) and age (40–64, ≥65 years old); model 2 for factors in model 1 plus BMI category (<18.5,18.5–23.9, ≥24–26.9, and 27 kg/m$^2$), smoking status (current, non-current smoker), exercise (regular, non-regular exercise), daily alcohol consumption (yes, no), menopause status (yes, no), marriage status (living with spouse, etc.), educational level (≥9 years, <9 years), and income [≥40,000 New Taiwan dollars (NTD), <40000 NTD]; and model 3 for factors in model 2 plus serum creatinine, alanine aminotransferase (ALT), hyperthyroidism, hypertension, hyperlipidemia, high calcium diet, long-term systemic steroid use, and oophorectomy.

Potential effect modifiers, such as sex and age (cutoff 65 years), were assessed using the likelihood ratio test to compare the goodness-of-fit of the models with and without the interaction terms in the fully adjusted model 3.

Sensitivity analysis was performed to test the robustness of our results, excluding events in the first year to enhance causal relationships and excluding fractures from the endpoint. To determine the correlation between glycemic control and osteoporosis risk, participants were also divided into two groups according to HbA1 level (cutoff 7%), after which Cox proportional hazards regression analysis was conducted. All analyses were performed using SAS version 9.4 (SAS Institute, Cary, NC) with a two-tailed alpha level of <0.05 indicating statistical significance.

## Results

### Descriptive analyses

After excluding 3375 participants at baseline who were pregnant within 1 year of the 2002 TwSHHH (n = 18), ≦40 years old (n = 3079), diagnosed with osteoporosis before the 2002 TwSHHH (n = 227), had missing data (n = 1), and were diagnosed with type 1 diabetes (n = 50), a total of 3331 participants [1690 men and 1641 women; mean (standard deviation) age of 55.1 (11.8) years] were included. The study flow chart is presented in Fig 1.

Baseline characteristics of the study population are detailed in Table 1. Overall, 390 individuals were identified to have had diabetes, yielding a prevalence rate of 11.7%. Expectedly participants with diabetes tended to be older (mean age 59.5 years old) and had significantly higher BMI, waist circumference, systolic BP, and diastolic BP compared to those without diabetes. The diabetes group had more participants living with their spouse as well as a higher education level, greater monthly income, and higher proportion of participants with menopause, long-term systemic steroid use, hypertension, and hyperlipidemia, compared to the

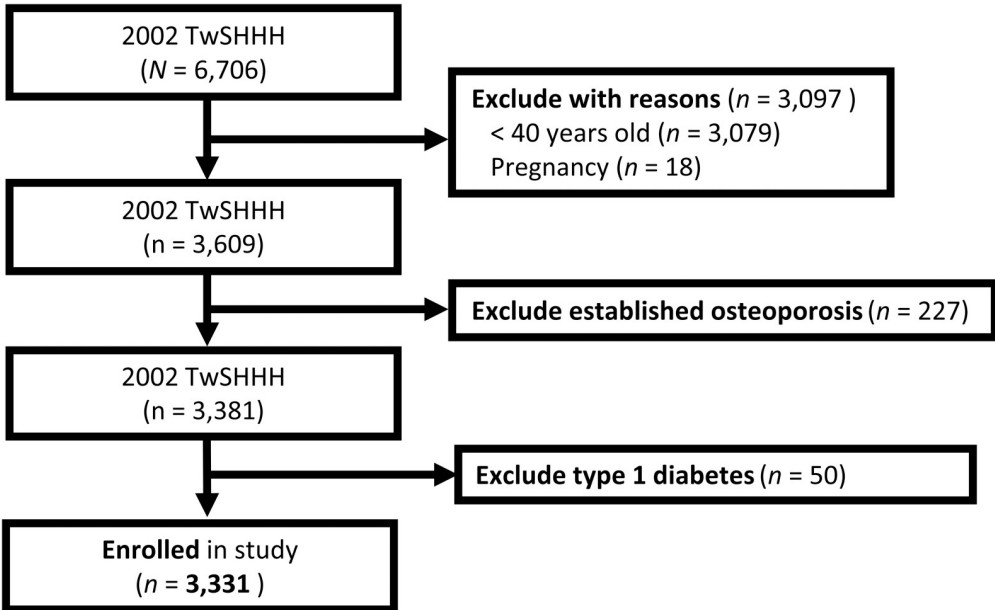

TwSHHH: Taiwanese Survey on Hypertension, Hyperglycemia, and Hyperlipidemia

**Fig 1. Flow diagram of participant enrollment.**

nondiabetes group. Diabetes was significantly associated with higher levels of FPG, triglyceride, low-density lipoprotein cholesterol, ALT, and creatinine and lower levels of high-density lipoprotein cholesterol. No difference in lifestyle factors, such as smoking status, daily alcohol consumption, and exercise habits, were observed between both groups.

Over a median follow-up duration of 13.6 years, 792 incident cases of osteoporosis were documented, with the diabetes and nondiabetes groups having an incidence rate of 34.20 and 20.55 per 1000 person-years, respectively.

Kaplan–Meier survival curves for osteoporosis according to the presence of diabetes showed that participants with diabetes had significantly lower osteoporosis event-free rates compared with those without diabetes (Fig 2).

### Inferential analyses

Table 2 shows the HRs and 95% CIs for osteoporosis according to the presence of diabetes. After adjusting for age and sex, the HR for osteoporosis in the diabetes group was 1.44 (95% CI: 1.18–1.75; $p < 0.001$). After additional adjustment for BMI, smoking status, daily alcohol consumption, regular exercise habits, menopause, marital status, educational level, and income, the HR for osteoporosis in the diabetes group was 1.37 (95% CI: 1.11–1.69; $p = 0.003$). After additional adjustment for serum creatinine level, ALT, hyperthyroidism, hypertension, hyperlipidemia, high calcium diet, long-term systemic steroid use, oophorectomy, the HR for osteoporosis in the diabetes group was 1.37 (95% CI: 1.11–1.69; $p = 0.004$).

Subgroup analysis according to age categories ($<65/\geq65$ years old) found that the association between the presence of diabetes and risk for osteoporosis was marginally modified by age, which suggested that diabetes plays a more significant role in the risk of osteoporosis among non-elderly individuals (interaction $p = 0.06$).

**Table 1. Baseline characteristics of the study participants.**

|  | Without diabetes (n = 2941) | With diabetes (n = 390) | *p* value |
|---|---|---|---|
|  | Mean (SD) | Mean (SD) |  |
| Age (years old) | 54.5 (11.8) | 59.5 (11) | <0.05* |
| BMI (kg/m$^2$) | 23.8 (3.3) | 25.2 (3.7) | <0.05* |
| Waist circumference (cm) | 82.3 (9.8) | 87.8 (10.3) | <0.05* |
| Systolic blood pressure (mmHg) | 120.9 (18.6) | 129.9 (19.2) | <0.05* |
| Diastolic blood pressure (mmHg) | 77.9 (11.5) | 80.4 (11.3) | <0.05* |
| Fasting plasma glucose (mg/dL) | 90.6 (9.7) | 157.8 (65.8) | <0.05* |
| Triglycerides (mg/dL) | 133.6 (82.5) | 189.1 (124.7) | <0.05* |
| LDL-C (mg/dL) | 122.1 (26.3) | 129.1 (31.6) | <0.05* |
| HDL-C (mg/dL) | 57.1 (15.8) | 52 (18.8) | <0.05* |
| ALT (mg/dL) | 20.3 (14.4) | 22.8 (19.0) | <0.05* |
| Creatinine (mg/dL) | 0.9 (0.3) | 1.0 (0.6) | <0.05* |
| HbA1C (%) | 5.2 (0.4) | 7.9 (1.9) | <0.05* |
|  | n (%) | n (%) |  |
| 40–64 (years) | 2310 (78.5) | 258 (66.2) | <0.05* |
| ≥ 65 (years) | 631 (21.5) | 132 (33.9) |  |
| Women | 1455 (49.5) | 186 (47.7) | 0.51 |
| BMI < 18.5 (kg/m$^2$) | 95 (3.5) | 10 (2.9) | <0.05* |
| 18.5 ≤ BMI < 24 kg/m$^2$ | 1432 (52.4) | 124 (35.6) |  |
| 24 ≤ BMI < 27 kg/m$^2$ | 818 (29.9) | 120 (34.5) |  |
| BMI ≥ 27 kg/m$^2$ | 388 (14.2) | 94 (27) |  |
| Current smokers | 691 (23.5) | 103 (26.4) | 0.20 |
| Alcohol use almost every day | 156 (5.3) | 19 (4.9) | 0.72 |
| Regular exercise habit | 776 (26.4) | 105 (26.9) | 0.82 |
| Living with spouse | 2369 (80.6) | 297 (76.2) | <0.05* |
| Educational level (≥9 years of schooling) | 1125 (38.3) | 89 (22.8) | <0.05* |
| Average month income ≥ 40,000 NTD | 644 (21.9) | 56 (14.4) | <0.05* |
| Menopause | 730 (24.8) | 148 (38) | <0.05* |
| Oophorectomy | 65 (2.2) | 8 (2.1) | 0.84 |
| High calcium diet | 1353 (46) | 164 (42.1) | 0.14 |
| Long-term systemic steroid use | 224 (7.6) | 41 (10.5) | <0.05* |
| Hypertension | 637 (21.7) | 148 (38) | <0.05* |
| Hyperlipidemia | 1271 (43.2) | 254 (65.1) | <0.05* |
| Hyperthyroidism | 619 (21.1) | 71 (18.2) | 0.75 |

ALT, Alanine Aminotransferase; BMI, Body mass index; HDL-C, High-density lipoprotein cholesterol; LDL-C, Low-density lipoprotein cholesterol; NTD, New Taiwan Dollars; SD: standard deviation

*$p < 0.05$.

No difference was found between subgroups stratified according to sex (interaction p = 0.97). After stratification, a significantly higher risk of osteoporosis was still observed among women (adjusted HR: 1.42, 95% CI: 1.09–1.85) but not men (adjusted HR: 1.34, 95% CI:0.94–1.92) (Table 3).

Sensitivity analyses demonstrated that results excluding events within 1 year after the 2002 TwSHHH were consistent with those obtained in the main analyses (Table 4).

Participants with HbA1C ≥7 had significantly higher osteoporosis risk than those with HbA1C <7 (adjusted HR:1.49, 95% CI: 1.15–1.92; *p* = 0.002) (Table 5).

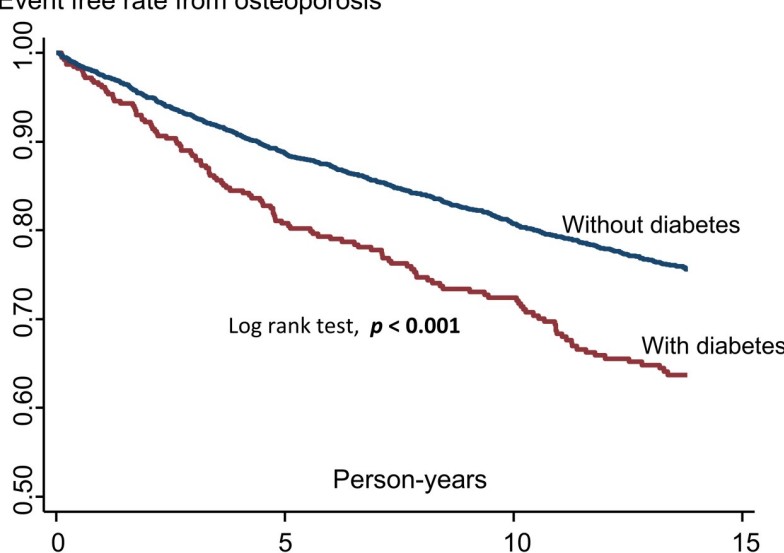

**Fig 2. Kaplan–Meier survival curves for osteoporosis according to the presence of diabetes.**

## Discussion

The current study suggested a positive association between the presence of T2D and incidences of osteoporosis in a Taiwanese population, with such an association being more pronounced among those aged <65 years old. This result indicated the presence of several important risk factors apart from T2D that should not be neglected among older individuals. Our results showed no difference on the basis of sex, suggesting that T2D may be associated with osteoporosis in both men and women. Moreover, the current study found that there was positive correlation between poor glucose control and osteoporosis risk.

Osteoporosis is clinically diagnosed using BMD or the presence of non-traumatic fractures. Although numerous studies have demonstrated an association between T2D and the risk of

**Table 2. Risk of osteoporosis according to the presence of diabetes.**

| Variables | With diabetes | Without diabetes |
|---|---|---|
| Participants | 390 | 2,941 |
| Person years | 3596.43 | 32,550.45 |
| Events | 123 | 669 |
| Incidence rate (per 1,000 person years) | 34.20 | 20.55 |
| Model 1 | 1.44 (1.18, 1.75)* | 1 |
| Model 2 | 1.37 (1.11, 1.69)* | 1 |
| Model 3 | 1.37 (1.11, 1.69)* | 1 |

Data are presented as hazard ratios with 95% confidence intervals.

Model 1: Adjusted for sex and age (40–64 and $\geq$65 years old).

Model 2: Additional adjustment for body mass index (<18, 18–24, 24–27, and $\geq$27kg/m$^2$), current smoking (yes/no), daily alcohol consumption (yes/no), regular exercise habit (yes/no), menopause (yes/no), living with spouse (yes/no), educational level (</$\geq$9 years of schooling), average monthly income (</$\geq$40,000 New Taiwan Dollars).

Model 3: Additional adjustment for serum creatinine level, alanine aminotransferase, hyperthyroidism, hypertension, hyperlipidemia, high calcium diet, long-term systemic steroid use, and oophorectomy.

**Table 3. Subgroup analyses for the risk of osteoporosis according to the presence of diabetes.**

| Variables | With diabetes | Without diabetes | $p_{interaction}$ |
|---|---|---|---|
| **Age** | | | 0.06 |
| 40–65 years old | 1.34 (0.94, 1.92) | 1 | |
| ≥65 years old | 1.08 (0.77, 1.51) | 1 | |
| **Sex** | | | 0.97 |
| Women | 1.42 (1.09, 1.85) | 1 | |
| Men | 1.34 (0.94, 1.92) | 1 | |

Adjustment based on model 3. Data are presented as hazard ratios and 95% confidence intervals.

**Table 4. Sensitivity analyses for the risk of osteoporosis according to the presence of diabetes.**

| Sensitivity analyses | With diabetes | Without diabetes |
|---|---|---|
| Excluding events in the first year | 1.39 (1.1, 1.74)* | 1 |
| Osteoporosis without fracture | 1.52 (1.11, 2.08)* | 1 |

Adjustment based on model 3. Data are presented as hazard ratios and 95% confidence intervals.

**Table 5. Risk of osteoporosis according to HbA1c level.**

| HbA1c level | HbA1c <7 | HbA1c ≥7 |
|---|---|---|
| HR | 1 | 1.49 (1.15, 1.92)* |

Adjustment based on model 3. Data are presented as hazard ratios and 95% confidence intervals.

osteoporotic fractures [10, 11, 14, 26–28], the results of previous studies on osteoporosis prevalence among patients with T2D are inconsistent. Accordingly, some studies showed that those with T2D had a lower, if not similar, of prevalence of osteoporosis determined using BMD compared to controls [29–31]. One possible explanation could be the degenerative changes and diffuse idiopathic skeletal hyperostosis frequently found in patients with T2D [32]. In contrast, a systemic review in China reported a higher prevalence of osteoporosis in patients with T2D [33]. A cross-sectional Korean population study also found an association between reduced BMD and diabetes duration [34]. Despite the lack of BMD data in our database, we attempted to use a composite diagnosis of osteoporosis through ICD9-CM codes of both osteoporosis and non-traumatic fractures and ATC codes of anti-osteoporotic agents to establish a more comprehensive osteoporosis diagnosis. Our data consistently suggested a significantly greater risk of osteoporosis among Taiwanese patients with T2D.

Long-term exposure to diabetes promotes changes in bone metabolism and impairs bone micro-architecture through multiple mechanisms, including elevated insulin levels, hypercalciuria, reduced renal function, obesity, more advanced glycation end products in collagen, angiopathies, neuropathies, and inflammation [7, 35]. In particular, insulin stimulates osteoblast proliferation and differentiation, whereas high glucose levels directly affect osteoblast metabolism and maturation by altering gene expression and diminishing bone mineral quality (7). Meanwhile, BMI is believed to be positively associated with BMD due to increased loading, adipokines, and higher aromatase activity [36]. Increased risk for falls caused by diabetic retinopathy, advanced cataracts, hypoglycemia, peripheral neuropathy, foot ulcers, polyuria, and

decreased reflexes may also increase the rate at which osteoporosis is diagnosed in patients with T2D [37].

The goal of osteoporosis treatment has always been fracture prevention. As such, apart from those diagnosed with osteoporotic fractures, patients diagnosed with osteoporosis without fracture should also be monitored. Our findings highlight the need for clinicians to monitor bone health, including BMD and fracture conditions, in patients with T2D. Early interventions aimed at enhancing bone health have to be implemented to prevent osteoporosis events, even in younger populations and regardless of sex.

This study has several strengths. First, although evidence of a higher prevalence of osteoporosis had been found in Asian populations, prospective cohort studies have not been available. To the best of our knowledge, this has been the first cohort study in Asia to explore the relationship between T2D and composite osteoporosis risk. Second, this study was a representative cohort study with validated outcomes based on the TwSHHH database. Moreover, the community-based populations from the TwSHHH database may reduce selection bias. The TwSHHH database includes important clinical laboratory variables and socioeconomic and lifestyle factors needed to adjust for potential confounding factors. Lastly, unlike most studies, our study surveyed younger populations, although marginal differences according to age had been noted.

Several potential limitations of the current study are worth noting. First, no BMD data for outcome evaluation had been available. To detect most events, both ICD9-CM codes for osteoporosis or osteoporotic fractures and ATC codes of anti-osteoporotic drugs determined from the NHIRD were used. Second, relatively few incident cases of osteoporosis had been observed for risk estimations. A median follow-up duration of 13.6 years already makes our result significant; however, osteoporosis may take longer to develop. If we follow a longer duration, more prominent results can be seen. Third, limited information was available regarding the effects of antidiabetic medications, calcium, or vitamin D on osteoporosis given that our study focused on primary prevention. Fourth, several of the covariates assessed only at baseline can change over time. Some individuals without T2D at baseline may eventually develop diabetes, whereas those diagnosed with T2D at baseline will have retained their diagnosis, which may be of particular significance. Considering the already significant positive result before adjustment, no further analysis was conducted on those who were newly diagnosed with diabetes.

## Conclusions

T2D was significantly associated with the risk of osteoporosis, especially among non-elderly participants, regardless of sex.

## Supporting information

**S1 Table. ICD-9-CM diagnostic codes for osteoporosis.**
(DOCX)

**S2 Table. Anatomical therapeutic chemical codes used to define the medications in the study cohort.**
(DOCX)

**S3 Table. The definition of the covariates in the study cohort.**
(DOCX)

**S1 File. Raw data.**
(DOCX)

## Acknowledgments

The authors would like to thank Enago (www.enago.tw), the editing brand of Crimson Interactive Pvt., Ltd. for the English language review.

## Author Contributions

**Conceptualization:** Hsin-Hui Lin, Hsin-Yin Hsu, Ming-Chieh Tsai, Tzu-Lin Yeh.

**Data curation:** Hsin-Hui Lin, Le-Yin Hsu, Kuo-Liong Chien, Tzu-Lin Yeh.

**Formal analysis:** Hsin-Hui Lin, Le-Yin Hsu, Tzu-Lin Yeh.

**Investigation:** Hsin-Hui Lin.

**Methodology:** Hsin-Yin Hsu, Ming-Chieh Tsai, Le-Yin Hsu, Kuo-Liong Chien, Tzu-Lin Yeh.

**Supervision:** Tzu-Lin Yeh.

**Writing – original draft:** Hsin-Hui Lin.

**Writing – review & editing:** Hsin-Yin Hsu, Tzu-Lin Yeh.

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
