## [Decision Letter · Decision Letter 0]

20 Apr 2021

PONE-D-21-09780

Association between type 2 diabetes and risk of osteoporosis: a nationwide cohort study

PLOS ONE

Dear Dr. Yeh,

Thank you for submitting your manuscript to PLOS ONE. After careful consideration, we feel that it has merit but does not fully meet PLOS ONE’s publication criteria as it currently stands. Therefore, we invite you to submit a revised version of the manuscript that addresses the points raised during the review process.

Reviewer 2 has raised important questions about the cohort, the method of identifying exposure, and endpoints that must be addressed.  Data availability must be addressed.  Reviewer 1 wishes clarification about how models 2 and 3 differ.

We look forward to receiving your revised manuscript.

Kind regards,

Robert Daniel Blank, MD, PhD

Academic Editor

PLOS ONE

Journal Requirements:

"Our protocol was reviewed and approved by the Research Ethics Committee of National Taiwan University Hospital. The committee was organized under and operates in accordance with the Good Clinical Practice Guidelines [NTUH-REC Number: 201901103W (Institutional Review Board reference)].

The data were analyzed anonymously.".   

Please provide additional details regarding participant consent. In the ethics statement in the Methods and online submission information, please ensure that you have specified what type you obtained (for instance, written or verbal, and if verbal, how it was documented and witnessed). If your study included minors, state whether you obtained consent from parents or guardians. If the need for consent was waived by the ethics committee, please include this information

4. Please include captions for ALL your Supporting Information files at the end of your manuscript, and update any in-text citations to match accordingly. Please see our Supporting Information guidelines for more information: http://journals.plos.org/plosone/s/supporting-information.

Reviewers' comments:

Reviewer's Responses to Questions

**Comments to the Author**

1. Is the manuscript technically sound, and do the data support the conclusions?

Reviewer #1: Partly

Reviewer #2: Partly

2. Has the statistical analysis been performed appropriately and rigorously? 

Reviewer #1: Yes

Reviewer #2: Yes

3. Have the authors made all data underlying the findings in their manuscript fully available?

Reviewer #1: Yes

Reviewer #2: No

4. Is the manuscript presented in an intelligible fashion and written in standard English?

Reviewer #1: Yes

Reviewer #2: Yes

5. Review Comments to the Author

Reviewer #1: Model 3 adjusts for oophorectomy. Unsure of added value given that menopause status was already adjusted for in Model 2.

The study collects HbA1c values. The authors could augment their study by looking the correlation of glycaemic control to risk of osteoporosis.

Reviewer #2: There is substantial evidence, giving rise to many meta-analyses, for increased risk of fracture in patients with type 2 diabetes (T2D) not explained from bone mineral density (BMD) which is paradoxically higher in T2D. The current report claims that “cohort studies estimating composite osteoporosis risk have been lacking”. The composite outcome proposed by the authors includes incident fractures, new diagnosis of osteoporosis (ICD-9-CM codes) or anti osteoporosis medications (NHIRD), but is problematic as noted below. Using a modest-sized cohort from a cardiovascular study (2002 Taiwanese Survey on Prevalence of Hypertension, Hyperglycemia, and Hyperlipidemia) the authors were able to identify 3,331 eligible participants including 390 with T2D at baseline. During median follow-up of 13.6 years, the authors noted an increased risk for the composite end point (adjusted HR 1.37) with perhaps slightly greater risk in younger individuals (HR 1.34 age 40 – 60 years, 1.08 age ≥ 65 years, p–interaction 0.06). The authors conclude that T2D was associated with increased risk for osteoporosis.

Major comments:

1. The composite end point chosen will conflate opposing effects seen in the individual end points. Fracture risk would be expected to be increased in T2D, whereas diagnosis and treatment of osteoporosis (from BMD) would be lower. The authors need to analyze and report the individual end points separately and, if they are divergent, reconsider the use of the composite measure.

2. The absence of BMD measurements limits the authors ability to draw conclusions about what triggers a diagnosis of osteoporosis. The use of osteoporosis diagnosis ICD-9-CM codes is an uncertain surrogate for BMD. Diagnosis of osteoporosis from BMD is usually dependent upon referral for screening DXA. Could more frequent medical contacts for T2D patients lead to more frequent DXA (differential ascertainment bias)?

3. From Table S1, the fracture end points are limited to vertebral (difficult to assess from ICD-9-CM codes), humerus and radio-ulnar. Hip fractures are the most strongly associated with T2D yet are not reported. Why?

Minor comments:

1. The title refers to a “nationwide” cohort study. This is incorrect since the cohort is only a small sample of the Taiwanese population.

2. The Introduction refers to risk factors for osteoporosis and “…multiple mechanisms involved, such as type 2 diabetes”. A reference is required. The Introduction refers to BMD loss (References #15, 16) but needs to highlight the abundant data showing greater baseline BMD in individuals with T2D.

3. Given the long follow-up many covariates assessed only at baseline will change over time. Some individuals without T2D at baseline will develop diabetes. Although the authors may not be able to address this in their analyses, it needs to be discussed.

4. Table S3 shows that oral corticosteroid use was defined from 2002-2015 data. This creates a situation where the exposure may actually follow the outcome. Time-varying methods or using a fixed covariate from baseline data avoids this.

5. The authors explain "The TwSHHH database contains data on the 2002 and 2007 TwSHHH populations linked together with the 2001 NIHS and NHIRD." It is unclear how the 2007 data were used, if at all, since all outcome measures seem to come from the NHIRD.

6. It is unclear why the sensitivity analysis excluding events within the first year was performed (Table 4). Please explain.

7. Duration alone does not overcome the many limitations. Please reword "duration of 13.6 years makes our results convincing."

8. The age stratification in Table 3 is probably incorrect (40 – 60 years, elsewhere < 65 years).

9. There are many grammatical errors that need to be corrected. Editing by a native English language speaker is required. The authors use sex and gender interchangeably. These are slightly different concepts and consistent language should be used.

6. PLOS authors have the option to publish the peer review history of their article (what does this mean?). If published, this will include your full peer review and any attached files.

Reviewer #1: No

Reviewer #2: No

---

## [Author Response · Author response to Decision Letter 0]

27 May 2021

Responses to each point raised by the academic editor:

Thank you for having taken your time to provide us with reminders and suggestions which allow us to improve the quality of our manuscript. We have carefully made revisions according to your reminders. 

Response: We appreciate for editor’s friendly reminder. We have corrected the format according to the PLOS ONE style templates including the capitalization of title, some level 2 heading in Methods and Results sections and the font size of table 1. We also adjust the file name of “raw code” to “S1 file” according to the style requirements. 

2. Thank you for including your ethics statement: "Our protocol was reviewed and approved by the Research Ethics Committee of National Taiwan University Hospital. The committee was organized under and operates in accordance with the Good Clinical Practice Guidelines [NTUH-REC Number: 201901103W (Institutional Review Board reference)].The data were analyzed anonymously.". Please provide additional details regarding participant consent. In the ethics statement in the Methods and online submission information, please ensure that you have specified what type you obtained (for instance, written or verbal, and if verbal, how it was documented and witnessed). If your study included minors, state whether you obtained consent from parents or guardians. If the need for consent was waived by the ethics committee, please include this information.

Response: Thanks for the kindly reminder. We added the information accordingly in the Methods section: “Written informed consent was obtained from all the participants.” (Page 6)

Response：Thanks for your advice. The datasets generated and analyzed during the current study are not publicly available due to the terms of consent to which the participants agreed but data are however available from the authors upon reasonable request and with permission of the Health Promotion Administration at the Ministry of Health and Welfare in Taiwan. We revised our cover letter and provided contact information for a data access committee, ethics committee, or other institutional body to which data requests may be sent. 

4. Please include captions for ALL your Supporting Information files at the end of your manuscript, and update any in-text citations to match accordingly. Please see our Supporting Information guidelines for more information: http://journals.plos.org/plosone/s/supporting-information.

Response: Thanks for your reminder. We added the raw data file at the end of our manuscript as “S1 File. Raw data.” (Page 22) as the guidelines. 

Response to each point raised by reviewer #1:

Thank you for having taken your time to provide us with valuable feedback and comments which allow us to improve the quality of our manuscript. We have carefully made revisions according to your comments and suggestions. 

1. Model 3 adjusts for oophorectomy. Unsure of added value given that menopause status was already adjusted for in Model 2.

Response: Thank you so much for your valuable comments. Oophorectomy is one of a well-known osteoporosis factor. According to our original questionnaire, oophorectomy may include partial oophorectomy. Partial oophorectomy may lead to early menopause. This condition is not included in menopause status when answering questionnaire. (E.K. Bjelland, P. Wilkosz, T.G. Tanbo, A. Eskild, Is unilateral oophorectomy associated with age at menopause? A population study (the HUNT2 Survey), Human Reproduction, Volume 29, Issue 4, April 2014, Pages 835–841, https://doi.org/10.1093/humrep/deu026, M. Rosendahl, M. K. Simonsen & J. J. Kjer (2017) The influence of unilateral oophorectomy on the age of menopause, Climacteric, 20:6, 540-544, DOI: 10.1080/13697137.2017.1369512)

2. The study collects HbA1c values. The authors could augment their study by looking the correlation of glycaemic control to risk of osteoporosis.

Response: Thank you so much for your valuable comments. We have analyzed the correlation of HbA1c and risk of osteoporosis as your opinion (Method, Page 8 and Results, Page 14, Discussion, Page 15). We divided the participants into 2 group according to HbA1c value. The group with HbA1C ≥7 has significantly higher osteoporosis risk than the group with HbA1C <7 (adjusted HR:1.46, 95% CI: 1.23-1.90). 

Response to reviewer #2:

Thank you for having taken your time to provide us with valuable feedback and comments which allow us to improve the quality of our manuscript. We have carefully made revisions according to your comments and suggestions. 

Major comments:

1. The composite end point chosen will conflate opposing effects seen in the individual end points. Fracture risk would be expected to be increased in T2D, whereas diagnosis and treatment of osteoporosis (from BMD) would be lower. The authors need to analyze and report the individual end points separately and, if they are divergent, reconsider the use of the composite measure.

Response: Thank you so much for your valuable comments. After stratification, a significantly higher osteoporosis risk was still observed among participants without fracture (adjusted HR: 1.52, 95% CI: 1.11–2.08) but not participants with fracture (adjusted HR :1.22, 95% CI:0.92–1.63). The end points are consistent in direction. We have added one more sensitivity analysis to clarify the question (Method, Page 8 and revised Table 4, Page 14). 

Table 4. Sensitivity analyses for the risk of osteoporosis according to the presence of diabetes 

 Sensitivity analyses With diabetes Without diabetes 

Excluding events in the first year 1.39 (1.1, 1.74)* 1 

Osteoporosis without fracture 1.52 (1.11, 2.08)* 1

Adjustment based on model 3, presented with hazard ratios and 95% confidence intervals 

2. The absence of BMD measurements limits the authors ability to draw conclusions about what triggers a diagnosis of osteoporosis. The use of osteoporosis diagnosis ICD-9-CM codes is an uncertain surrogate for BMD. Diagnosis of osteoporosis from BMD is usually dependent upon referral for screening DXA. Could more frequent medical contacts for T2D patients lead to more frequent DXA (differential ascertainment bias)?

Response: Thank you so much for your valuable comments. More frequent medical contacts for T2D patients may introduce possible bias. However, in Taiwan, DXA is a self-paid exam, so T2D patient would not be advised to do DXA in current routine care. Also, accessibility to medical care is high in Taiwan, DXA is easily arranged for non T2D patients. Most osteoporosis diagnosis were made during health exam or after fracture. Therefore, the effect was supposed to be small. 

3. From Table S1, the fracture end points are limited to vertebral (difficult to assess from ICD-9-CM codes), humerus and radio-ulnar. Hip fractures are the most strongly associated with T2D yet are not reported. Why?

Response: Thank you so much for your valuable comments. We have added back the missing ICD-9-CM codes for femoral neck fractures in our revised Table S1 as below. 

Femoral neck fractures 820, 820.0, 820.2, 820.8, 820.0x, 820.2x

Minor comments:

1. The title refers to a “nationwide” cohort study. This is incorrect since the cohort is only a small sample of the Taiwanese population.

Response: Thank you so much for your valuable comments. We have revised our title to “A Representative Cohort Study in Taiwan” and also deleted “nationwide” in abstract. (Abstract, Page 2) 

2. The Introduction refers to risk factors for osteoporosis and “…multiple mechanisms involved, such as type 2 diabetes”. A reference is required. The Introduction refers to BMD loss (References #15, 16) but needs to highlight the abundant data showing greater baseline BMD in individuals with T2D.

Response: Thank you so much for your valuable comments. We have added references for “…multiple mechanisms involved, such as type 2 diabetes” on page 3. We have revised “While most studies agree that diabetes increases fracture risk,(10-14), the association between T2D and risk for bone mineral density (BMD) loss in prior studies have been inconsistent.(15, 16) “ to “Although most studies agree that diabetes increases fracture risk (10-14), the association between T2D and risk for bone mineral density (BMD) loss have been inconsistent in prior studies, with abundant data showing a greater baseline BMD in individuals with T2D (15-17).” (Page 3)

3. Given the long follow-up many covariates assessed only at baseline will change over time. Some individuals without T2D at baseline will develop diabetes. Although the authors may not be able to address this in their analyses, it needs to be discussed.

Response: Thank you for your valuable comment. The control group may develop diabetes during follow-up period, but T2D patient at baseline won’t become normal. The significance may be more prominent. Due to already significant positive result before adjustment, we did not further analysis the newly develop diabetes condition. We have added the discussion of the limitation of our statistic method as “Fourth, several of the covariates assessed only at baseline can change over time. Some individuals without T2D at baseline may eventually develop diabetes, whereas those diagnosed with T2D at baseline will have retained their diagnosis, which may be of particular significance. Considering the already significant positive result before adjustment, no further analysis was conducted on those who were newly diagnosed with diabetes.” (Discussion, Page 17)

4. Table S3 shows that oral corticosteroid use was defined from 2002-2015 data. This creates a situation where the exposure may actually follow the outcome. Time-varying methods or using a fixed covariate from baseline data avoids this.

Response: Thank you for your valuable comment. According to previous studies, glucocorticoids related osteoporosis need long term steroid use and is reversible if stop steroid. So baseline data may not reflect the possible effect. We have excluded the oral corticosteroid use after event day (revised S3 table) and reanalysis. The new descriptive data is shown in Table 1 (Page 11). There is no change in other results after change the definition of oral corticosteroid use. 

Long-term systemic steroid use  224 (7.6) 41 (10.5) <0.05* 

5. The authors explain "The TwSHHH database contains data on the 2002 and 2007 TwSHHH populations linked together with the 2001 NIHS and NHIRD." It is unclear how the 2007 data were used, if at all, since all outcome measures seem to come from the NHIRD.

Response: Thank you for your valuable comment. We did not use 2007 data in the research this time. We have removed the statement about 2007 TwSHHH (Page 5).

6. It is unclear why the sensitivity analysis excluding events within the first year was performed (Table 4). Please explain.

Response: Thank you so much for your valuable comments. We excluded events within the first year to enhance causal relationship. The development of osteoporosis takes more than one year. We want to remove the possible bias as we can to make the causal relationship of T2D and osteoporosis stronger. We have added “to enhance causal relationships” in Method, page 8.

7. Duration alone does not overcome the many limitations. Please reword "duration of 13.6 years makes our results convincing."

Response: Thank you so much for your valuable comments. We have revised “Although osteoporosis may take longer to develop, a median follow-up duration of 13.6 years makes our results convincing.“ to “A median follow-up duration of 13.6 years already makes our result significant; however, osteoporosis may take longer to develop. If we follow a longer duration, more prominent results can be seen.” (Page 17)

8. The age stratification in Table 3 is probably incorrect (40 – 60 years, elsewhere < 65 years).

Response: Thank you so much for your valuable comments. We have revised the mistake in Table 3. 

Table 3. Subgroup analyses for the risk of osteoporosis according to the presence of diabetes 

 Variables With diabetes Without diabetes pinteraction 

Age 0.06 

  40–65 years old 1.34 (0.94, 1.92) 1 

  ≥65 years old 1.08 (0.77, 1.51) 1 

Sex 0.97 

  Women 1.42 (1.09, 1.85) 1 

  Men 1.34 (0.94, 1.92) 1 

9. There are many grammatical errors that need to be corrected. Editing by a native English language speaker is required. The authors use sex and gender interchangeably. These are slightly different concepts and consistent language should be used.

Response: Thank you so much for your valuable comments. We have edited the grammatical errors and inconsistent language according to a native English language speaker’s suggestion.

---

## [Decision Letter · Decision Letter 1]

28 Jun 2021

Association between type 2 diabetes and osteoporosis risk: a representative cohort study in Taiwan

PONE-D-21-09780R1

Dear Dr. Yeh,

We’re pleased to inform you that your manuscript has been judged scientifically suitable for publication and will be formally accepted for publication once it meets all outstanding technical requirements.

Kind regards,

Robert Daniel Blank, MD, PhD

Academic Editor

PLOS ONE

Additional Editor Comments (optional):

Reviewers' comments:

Reviewer's Responses to Questions

**Comments to the Author**

1. If the authors have adequately addressed your comments raised in a previous round of review and you feel that this manuscript is now acceptable for publication, you may indicate that here to bypass the “Comments to the Author” section, enter your conflict of interest statement in the “Confidential to Editor” section, and submit your "Accept" recommendation.

Reviewer #1: All comments have been addressed

2. Is the manuscript technically sound, and do the data support the conclusions?

Reviewer #1: Yes

3. Has the statistical analysis been performed appropriately and rigorously? 

Reviewer #1: Yes

4. Have the authors made all data underlying the findings in their manuscript fully available?

Reviewer #1: Yes

5. Is the manuscript presented in an intelligible fashion and written in standard English?

Reviewer #1: Yes

6. Review Comments to the Author

Reviewer #1: (No Response)

7. PLOS authors have the option to publish the peer review history of their article (what does this mean?). If published, this will include your full peer review and any attached files.

Reviewer #1: No

---

## [Editor Report · Acceptance letter]

5 Jul 2021

PONE-D-21-09780R1 

Association between type 2 diabetes and osteoporosis risk: a representative cohort study in Taiwan 

Dear Dr. Yeh:

I'm pleased to inform you that your manuscript has been deemed suitable for publication in PLOS ONE. Congratulations! Your manuscript is now with our production department. 

Kind regards, 

on behalf of

Professor Robert Daniel Blank 

Academic Editor

PLOS ONE